# Chatter Detection in Milling of Carbon Fiber-Reinforced Composites by Improved Hilbert–Huang Transform and Recurrence Quantification Analysis

**DOI:** 10.3390/ma13184105

**Published:** 2020-09-16

**Authors:** Rafał Rusinek, Paweł Lajmert

**Affiliations:** 1Faculty of Mechanical Engineering, Lublin University of Technology, 36 Nadbystrzycka Str, 20-618 Lublin, Poland; 2Faculty of Mechanical Engineering, Lodz University of Technology, 1/15 Stefanowskiego Str, 94-040 Łódź, Poland; pawel.lajmert@p.lodz.pl

**Keywords:** milling, chatter vibrations, time-frequency analysis, recurrence quantification analysis, CFRP

## Abstract

In the paper, the problem of chatter vibration detection in the milling process of carbon fiber-reinforced plastic is investigated. Chatter analysis may be considered theoretically based on data from impact test of an end mill cutter. However, a stability region obtained in such way may not agree with the real one. Therefore, this paper presents a method that can predict chatter vibrations based on cutting force components measurements. At the beginning, a stability lobe diagram is created to establish the range of experimental test in the plane of tool rotational speed and depth of cut. Next, an experiment of composite milling is performed. The experimentally-measured time series of cutting forces are decomposed with the use of the improved Hilbert–Huang transform (HHT). To detect chatter, statistical methods and recurrence quantification analysis (RQA) are used. However, much better results are obtained when new chatter indexes are proposed. The indexes, derived directly from the HHT and RQA methods, can be used to build an effective chatter prediction system.

## 1. Introduction

Carbon fiber-reinforced composites (CFRC) are becoming more and more popular every year, year-round, as a material for the production of responsible machine parts in many industries such as aerospace, marine and automotive. These materials are featured by high specific strength and stiffness at low weight [1]. Due to the lower weight of the final product, they lead to less energy and fuel consumption or better operational productivity. Although the products can be made in complex shapes, they need however some machining, e.g., milling process, to achieve final dimensional and assembly requirements. The features of these materials that make them a perfect choice in a variety of demanding applications, generate, however, many difficulties in their machining and choosing the right cutting tool. The key problem in machining CFRC is the choice of optimal machining parameters due the required product quality, abrasive wear of the cutting tool or the occurrence of so-called chatter vibrations, i.e., strong relative vibrations between a tool and a machined workpiece. The high level of vibrations usually adversely influences the surface quality parameters, but can also contribute to accelerated tool wear or even tool or the machined part damage. These difficulties can be further intensified by the high compliance of typical parts made of these materials. Furthermore, a need to increase productivity and product quality causes the material removal rate to be increased. Higher machining efficiency can be attained by applying high speed or high-efficiency machining, which may, however, intensify the risk of chatter vibrations.

Therefore, the primary task of the milling machine operator is to select adequate machining conditions that will protect the process from the adverse effects of chatter vibrations, while ensuring the maximum possible economics or machining efficiency. Usually, stability lobes diagrams (SLDs) are created before machining, where the boundary between stable and unstable machining is defined in the domain of cutting tool rotational speed and depth of cut [2,3]. Various models are proposed, e.g., artificial neural networks [4], to describe a stable machining area and predict machining chatter. In practice, the area of stable machining may be different than expected, due to simplifications regarding model linearity and inaccuracy in identifying model parameters. As a result, process supervision systems are crucial, because they can diagnose the process state on-line and accurately modify the machining conditions when some symptoms of process instability appear [3,5,6].

There are many articles regarding the cutting of composite materials where different methods of avoiding problems with fiber pullout, fragmentation or delamination are presented [7,8,9,10,11]. Avoiding these problems is important, because they negatively affect the strength and durability of composite products. Moreover, experimental research of various cutting tools and their wear are presented in many papers, e.g., [12,13]. In addition, the influence of cutting parameters on cutting results and forces is studied [7,9,14]. A comprehensive review of problems with tool wear and final product quality, related to machining of composite materials, are presented in [15,16,17,18]. Fewer articles deal with the problem of chatter vibrations identification in the milling of composite materials [14,19,20].

Regenerative chatter, known also as the secondary one, is the most common type of chatter vibrations in machining [21]. However, in cutting composite materials, frictional, known as primary chatter [22,23,24], may also be a very important source of vibrations, because composite materials are very abrasive and have a variable friction coefficient. In general, the coefficient of friction grows as the cutting speed increases for all composite materials [25]. For this reason, cutting tools made of high-speed steel or carbide are often covered by special coatings, such as diamond-like carbon (DLC), that significantly reduce friction and extend the tool life. The problem of frictional chatter and chaotic vibrations, caused by dry friction are described among other effects in [21,26,27].

In practice, chatter vibrations may be suppressed using vibration eliminators, which change the dynamical characteristics of the machine-tool system or a phase between the external and internal modulations in trace regeneration [28,29]. However, these methods are passive. This means that a system that eliminates chatter should be tuned before each cutting operation. The effect of trace regeneration may be also reduced or canceled by a change of the phase shift using a variable spindle rotational speed [28]. During machining using multi-point tools stability may be increased by changing the rake angle or by using unequal spacing of cutting inserts in the milling head. A comprehensive study on the performance of variable helix and variable pitch end mills is provided in [30]. In this work, an analytical model of the cutting process for the prediction of chatter stability of variable helix end mills is also presented. However, solutions that use special tools increase production costs. Another method, widely used in industry, is the use of commercial applications, such as CutPro software, which allow us to create SLDs [31]. This software creates SLDs based on modal analysis of a tool–spindle system but its accuracy depends a lot on the number of modes taken into account and many other assumptions [32].

Due to the difficulties and limitations mentioned above many works concentrates on the development of methods that could detect the initial symptoms of chatter, based on measurements of different signals, i.e., force, vibrations or acoustic emission [14,19,33,34]. These signals are analyzed with the use of various signal processing techniques in time, frequency and time–frequency domains. In literature, works can be found that use, for this purpose, among others, standard and nonstandard statistical methods [5,14,35,36], entropy measures [5,19,36], wavelet transform [37,38,39], different versions of Hilbert–Huang transform (HHT) [5,36,40,41], recurrence plot technique (RP) and recurrence quantification analysis (RQA) [36,42]. Methods based on statistical and entropy measures are very popular in chatter analysis. They are used in grinding [43,44,45], turning [46] and milling processes [47]. Although nonstandard statistical methods are not new, they are often very effective in diagnostics of various machining processes. The methods of process state analysis mentioned here are presented in [48,49]. In these studies, chatter vibrations are investigated with the use of rescaled range and detrended fluctuation analysis. The RP technique is used to identify various physical phenomena [50], also among other in machining processes [14,19,35,36,42]. The use of HHT transform may also be a useful method for identification of early symptoms of process instabilities in different machining processes, e.g., in grinding [44] or milling [5,37,40,41]. However, still some methods of signal analysis are necessary to reveal some useful features from frequency components obtained after signal decomposition. A quite new method for signal decomposition and dynamical behavior analysis is presented in [51], where chatter is investigated using the Hilbert vibration decomposition (HVD) [52]. 

To conclude, in our opinion it is better when diagnosing the process state during the machining and avoid chatter vibrations by an adequate change of cutting conditions, e.g., cutting speed, with the highest possible machining efficiency at the same time. In the paper, investigations of the milling process stability of carbon fiber reinforced plastic (CFRP) based on force time series analysis are presented. In order to distinguish the chatter-free and chatter vibrations, the improved HHT transform and the RQA analysis are used. 

The paper is divided as follows. In Section 1, a brief review of the literature related to chatter vibrations is presented. Section 2 describes the experimental test stand, the methodology of experiments and numerical tools used for preliminary stability region analysis. Section 3 includes results of preliminary force measurements and a theoretical description of the methods used for the cutting force decomposition and signal analysis, i.e., the improved Hilbert-Huang transform and recurrence quantification analysis. In Section 4 results of force signal decomposition are presented. In Section 5, an analysis of force–frequency components is performed for identifying process instability. New indexes of process instability are introduced based on improved HHT and RQA methods. Next, some remarks regarding the real area of stable machining conditions are presented Finally, in Section 5, practical conclusions and propositions of future research are presented. 

## 2. Experimental Test Stand and Research Procedure 

The milling test of the CRFP workpiece was performed on the milling machine—BlueBird MG6037PKK (Certus, Wadowice, Poland, www.mg-certus.pl). During the experiment, two components of the total cutting force *F_x_*, *F_y_* were measured with the help of the experimental setup presented schematically in Figure 1.

The experimental setup was composed of two subsystems: a dynamometer for measuring cutting force components and a modal analysis system. The force measurement subsystem consisted of the piezoelectric dynamometer Kistler 9257B, the charge amplifier 5017B, the simultaneously-sampled differential amplifier NI SC2040 and the NI data acquisition card. Measuring cutting force components and storage in computer memory was the main aim of the subsystem. The second subsystem, i.e., the modal analysis system was used at the beginning to identify the dynamic characteristics of a tool-holder structure. It was composed of the modal hummer PCB 086C03, the ceramic shear accelerometer PCB 352B10 and the signal acquisition module NI 9234. Both subsystems of the experimental rig were integrated into a PC computer and controlled by CutPro9 and specially developed measurement software.

### 2.1. Tool-Holder Modal Analysis

At the beginning of the experiment, a single point impact test was done to determine the natural frequency, stiffness and damping ratio of the spindle–tool system, which are demanded to predict the stability lobes diagram (SLD). The modal hammer was applied to excite the tooltip. Next, the output signal was measured with the accelerometer fixed to the tooltip. After that, the modal parameters for the *X* and *Y* directions were obtained in the form of the frequency response function (FRF) presented in Figure 2. 

The first natural frequency of the tool-holder system was about 600 Hz, and the second one about 700 Hz. This can estimate chatter vibration frequency *f_o_* equal to about 600–700 Hz. The higher frequencies, about 1200 Hz shown in the FRF, were the second harmonic of chatter. Therefore, the SLD created by means of the CutPro9 software has the mother lobe, i.e., the first at the highest rotational speed *n_o_*, equal to *60f_o_*/*z*, where *z* is the number of the tool flutes (*z* = 2). According to the linear theory implemented also in CutPro9 software [2,53], the next lobes appear at *n = n_o_*/*l_o_* where *l_o_* is a lobe order, *l_o_ = 2*, *3*, etc. 

Based on the FRF function (Figure 2), a theoretical SLD was generated (Figure 3) by means of CutPro9 software for the feed rate per tooth *f_z_* = 0.05 mm and the radial depth of cut *a_e_* = 12 mm. However, to verify the SLD and to find new indexes useful in chatter prediction, machining conditions are proposed in the next subsection. 

### 2.2. Conditions of Experiment

As mentioned before, the theoretical SLD, obtained with the use of CutPro9 software, is shown in Figure 3. According to Figure 3, the experimental test was divided into two parts. First, the rotational speed *n* was changed from 2000 rpm to 8000 rpm, while the depth of cut *a_p_* equal to 0.5 mm. Next, in the second part of the test, rotational speed was constant and equal to 6000 rpm, while the cutting depth *a_p_* was changed from 0.5 to 2 mm. The feed rate per tooth *f_z_* and the radial depth of cut *a_e_* for all tests were constant and equal to *f_z_* = 0.05 mm and *a_e_* = 12 mm, as assumed for the SLD.

The range of parameters used during tests is collected in Table 1. The milling cutter was made of diamond-coated cutting steel, having a diameter of 12 mm and two flutes (*z* = 2). The experiment was repeated for every set of parameters applying identical conditions, that are: the tool and the workpiece position, tool wear and temperature of the test. Finally, 16 tests were performed and only the representative part of the time series was taken for the analysis.

*X* and *Y* components of the total cutting force were measured using a sampling frequency of 4 kHz. The analyses presented in the next section were undertaken to verify whether the theoretical stability region presented in Figure 3 was determined properly and to find some new features of measured signals useful in chatter vibrations prediction. 

## 3. Nonlinear Time Series Analysis

To predict stability and to identify chatter vibrations in the milling process, the analysis of the force-time series is shown in this section. Generally, in the cutting process, the measured cutting force components are usually non-stationary and nonlinear by nature. Therefore, it is necessary to use an adequate signal processing techniques which could properly decompose the analyzed signal into the specific frequency components and reveal the dynamic content of the signal. Standard methods like the short-time Fourier Transform or the wavelet transform (WT) can analyze stationary and non-stationary signals, but not nonlinear ones. When using WT the results rely to a great extent on the basic wavelet function and the discretization of scales. Improper selection of any of these parameters may reduce the applicability of this method in analyzing non-stationary and nonlinear signals. 

The HHT is a self-adaptive signal decomposition technique designed to analyze nonlinear signals [54]. This method can decompose the signal into separate frequency components for which instantaneous amplitude and frequency can be determined. The use of this technique should enable to solve process stability prediction problem. 

In order to verify the theoretical SLD the resultant cutting force *F_xy_* acting on the milling cutter in the XY plane, is taken into consideration. During preliminary force components analysis, it was found that the character of the *F_x_* and *F_y_* components is similar and they are shifted in phase by about 90 degrees. For this reason, the resultant cutting force *F_xy_* is considered to be a representative of milling dynamics. This force is calculated as the square root of the sum of force component *F_x_* and *F_y_* according to the following formula:(1)$$F_{xy}=\sqrt{F_{x}^{2}+F_{y}^{2}}.$$

As an example, changes of *F_x_*, *F_y_* components and resultant force *F_xy_* are presented in Figure 4 for stable and unstable conditions, that are taken from the SLD shown in Figure 3. The determination of the stability region seems to be quite easy based on the stability lobe diagram of CFRP. However, when visually analyzing changes of force components and resultant force, no clear differences between stable and unstable conditions can be discerned. As may be seen in Figure 4 amplitudes of resultant force are almost the same for stable and unstable conditions, although the waveforms obviously differ in the content of dynamic components. Therefore, the theoretical SLD of CFRP will be verified using the methods described in this section.

### 3.1. Improved Hilbert-Huang Transform

The HHT is a signal decomposition technique, introduced by Huang et al. [54], which recently has gained much attention due to its successful use in many applications. This method was developed especially for analysis of non-stationary and nonlinear signals changing even within a single oscillation cycle. In general, the HHT is composed of two processing methods, i.e., the empirical mode decomposition (EMD) and the Hilbert transform. The aim of the EMD is a decomposition of the signal *x(t)* into a set of so-called intrinsic mode functions (IMFs), denoted here as *c_j_*(*t*), *j = 1*, *...*, *m*−*1*. Assuming that the last IMF is a data trend *r_m_*(*t*), the original signal *x*(*t*) may be represented as follows:(2)$$x\left(t\right)=\overset{m-1}{\underset{j=1}{\sum }}c_{j}\left(t\right)+\;r_{m}\left(t\right).$$

The presented modes can have a variable amplitude and frequency along the time, unlike simple harmonic functions. By default, this method decomposes the signal in a few steps. The first finds all local minima and maxima points of the analyzed signal. These points are interpolated by cubic splines to create the upper and lower envelops, see Figure 5a. In the next step, a mean line *m*_1_(*t*) between the upper and lower envelopes is created and subtracted from the original signal *x*(*t*). As an effect, a new signal *h*_1_(*t*) is obtained, as presented in Figure 5b.
(3)$$h_{1}\left(t\right)=x\left(t\right)-m_{1}\left(t\right).$$

The above procedure is called a sifting process. It is repeated for the successive data *h_k_*(*t*) until the mean line between upper and lower envelopes is close enough to zero at any point.
(4)$$h_{k}\left(t\right)=h_{k-1}\left(t\right)-m_{k}\left(t\right),\;\;k=2,\;\ldots ,\;n.$$

At this moment, the data *h_n_*(*t*) becomes the first IMF component *c*_1_(*t*). In Figure 5c the data after the second sifting process are shown.
(5)$$c_{1}\left(t\right)=h_{n}\left(t\right).$$

This component represents the highest frequency component of the original signal *x*(*t*). Next, the first IMF component *c*_1_(*t*) is subtracted from the original signal *x*(*t*) and the whole sifting process is repeated for the new data *x*_1_(*t*).
(6)$$x_{1}\left(t\right)=x\left(t\right)-c_{1}\left(t\right)$$

In Figure 5d, new data *x*_1_(*t*) are shown. Thanks to this procedure, the successive components *c_j_*(*t*) of decreasing mean frequency are obtained. This process is stopped at the moment when the component *c_m_*(*t*) becomes small enough or it remains a monotonic function and no more IMFs can be extracted. As can be seen in Figure 5d, using the standard method proposed by Huang, the components are not properly separated after the two sifting operations and are partially mixed with each other.

Although the empirical mode decomposition has been shown to be very effective in many practical applications, it has some significant drawbacks. One of these drawbacks is the inability to separate components that are close enough to each other in the frequency domain [55]. In literature, this problem is called the separation problem. It also applies to the situation where the ratio of component amplitudes is appropriately small. In this case, the higher frequency component cannot be distinguished, as it does not form local extremes against the carrier component, i.e., the lower frequency component. As shown in Figure 5a, not all necessary interpolation points have been found, as they are not local extremes. This creates the possibility that separated components may be mixed, i.e., the higher frequency component may also partially contain a carrier component of lower frequency, see Figure 5c. This leads to the so-called mode mixing problem and to distortion of both separated components. Therefore, the sifting process proposed originally by Huang does not guarantee that all the necessary interpolation points will be found. A number of alternative methods of finding interpolation points for the upper and lower envelopes or finding the mean line have been proposed in the literature [55,56,57]. The sifting process may be improved by the use of a time-varying filter technique [55], a heuristic search optimization approach [56] or a local integral mean-based sifting [57]. Interpolation points may also be found based on a derivative of the analyzed signal. For the first derivative of a single component signal, the interpolation points for upper and lower envelopes correspond to the intersection of this derivative with the zero axis. For multi-component signals, the derivative of the whole signal contains the derivatives of the fast and low oscillating components. For this reason, in the paper [58] a number of sifting operations on the first derivative are done to obtain derivative of only the highest frequency component. The zero-crossing points of this derivative are used to find interpolation points for upper and lower envelopes. This method requires however many sifting operations and therefore may be time-consuming.

On the other hand, the mean line can be estimated directly from the inflection points derived based on local extremes of the first derivative of the signal, as shown in Figure 5e. In this case, inflection points correspond to the location of extremes of the odd derivative of the signal *x*(*t*). As can be seen in Figure 5f, after the first sifting operation, the high oscillating component was almost properly separated, although it requires several additional sifting operations in order to be isolated. In Figure 5g this higher frequency component is shown after two sifting operations. Since using this method, the sifting process may gradually distort the extracted component, in practice the sifting process should be performed once or at most twice. After that, the sifting process is carried out using the standard method proposed by Huang. In Figure 5h the second, low oscillating component is shown. As may be seen, both components were separated in a better way and no mode mixing occurred. Additionally, the presented method enables the separation of the signal components in a smaller number of sifting operations. This method, denoted here as DHHT (Derivative based HHT), is used in this work to decompose the resultant force *F_xy_* signal into individual frequency components. 

The second part of the HHT needs a Hilbert spectral analysis [59] that is engaged to each IMF component separately in order to calculate instantaneous amplitude and frequency. Based on the Hilbert transform the instantaneous amplitude and frequency obtained from the IMFs may be converted to full time–frequency distribution of energy contained in the data, i.e., a Hilbert spectrum [60]. Using the Hilbert transform, every signal *x*(*t*) may be altered into a complex function *z*(*t*) by including a complex part *y*(*t*) which is the same as *x*(*t*) but moved in phase by 90 degrees.
(7)$$z\left(t\right)=x\left(t\right)+jy\left(t\right)=a\left(t\right)e^{i\theta \left(t\right)}$$
where *a*(*t*) is the instantaneous amplitude and *θ*(*t*) is the phase of the analyzed signal. The instantaneous amplitude and frequency can be calculated based on *x*(*t*) and *y*(*t*) signals using the following formulas:(8)$$a\left(t\right)=\sqrt{x{\left(t\right)}^{2}+y{\left(t\right)}^{2}}$$
(9)$$\omega \left(t\right)=\;\dot{\theta }\left(t\right)=\frac{d}{dt}\left(tan^{-1}\left(\frac{y\left(t\right)}{x\left(t\right)}\right)\right).$$

Since the instantaneous frequency and amplitude cannot be used to describe a multi-component signal, the EMD and the Hilbert transform must be applied together. In the beginning, the original signal was transformed into separate frequency components and next, instantaneous amplitude and frequency were calculated.

### 3.2. Reccurence Quantificatin Analysis

After the DHHT analysis, several components of the force signals *F_xy_* are obtained, whichmay be responsible for chatter in milling. Here, these components (modes) were analyzed by means of the recurrence quantification analysis (RQA), which is based on the recurrence plots (RP) technique proposed by Eckmann [61]. The concept of the RP and RQA assumes that any time series can be presented as a delayed vector, with delay *d* in an *m*-dimensional space. Parameters *m* and *d* are called as the embedding dimension and the time delay, respectively. In the beginning, the embedding parameters (*m* and *d*) should be estimated. Here, the average mutual information function and the false nearest neighbors method is used to calculate embedding parameters. Having these parameters, the RP and the RQA can be done. The RP indicates all the time instants when the phase space trajectory of the dynamical system meets almost the same area in the phase space [50,62]. Mathematically, the RP is expressed as a matrix:(10)$$M_{ij}=\theta \left(\varepsilon -\left|s_{i}-s_{j}\right|\right),$$
where *θ* is the Heaviside step function, *ε* is a threshold parameter, **s_i_** and **s_j_** are delay vectors. If recurrence appears then **M_ij_** = 1, otherwise **M_ij_** = 0. These results are plotted in RPs as black (**M_ij_** = 1) and white (**M_ij_** = 0) dots, respectively. However, analyzing patterns in RPs is not objective enough therefore usually the RQA is used to describe RPs statistically. In this paper, the following RQA indicators are employed:The largest Lyapunov exponent (*L_yap_*) calculated by the Kantz algorithm described in [63],The averaged diagonal line length (*L*):
(11)$$L=\frac{\sum_{l=lmin}^{N}lP\left(l\right)}{\sum_{l=lmin}^{N}P\left(l\right)}.$$

The longest diagonal line length (*L_max_*):

(12)$$L_{max}=max\left(\left\{l_{i};i=1,\;\ldots ,N_{l}\right\}\right).$$

The L-entropy (*L_ent_*) is Shannon’s entropy of diagonal line segment distribution:

(13)$$L_{ent}=-\sum_{l=lmin}^{N}P\left(l\right)\;lnP\left(l\right).$$

The trapping time (*TT*) is the average length of the vertical lines:

(14)$$TT=\frac{\sum_{v=v_{min}}^{N}vP\left(v\right)}{\sum_{v=v_{min}}^{N}P\left(v\right)},$$
where *P*(*l*) is the histogram of the lengths *l* of the diagonal lines. *P*(*v*) is the histogram of the vertical lines lengths *v*, and *N* denotes the quantity of points on the phase space trajectory.

To calculate RQA indicators, the Tisean software was engaged [64]. These methods are commonly applied in the literature to analyze different signals from cutting processes [19,35,36,42,65] and to medical problems [66]. However, in the milling process, the conventional RQA indicators are not precise enough to recognize chatter, therefore, the improved chatter indicators (ICI) are proposed, defined as follows: *L_yap_* chatter index (*L_yap_CI*)
(15)$$L_{yap}CI=\frac{L_{yap}}{A_{DHHT}}$$

*L_max_* chatter index (*L_max_CI*)

(16)$$L_{max}CI=\frac{L_{max}}{A_{DHHT}}$$

*L_ent_* chatter index (*L_ent_CI*)

(17)$$L_{ent}CI=\frac{L_{ent}}{A_{DHHT}}$$

*L* chatter index (*LCI*)

(18)$$LCI=\frac{L}{A_{DHHT}}$$

*TT* chatter index (*TTCI*)

(19)$$TTCI=\frac{TT}{A_{DHHT}},$$
where *A_DHHT_* means the amplitude of the chatter force component calculated by means of the DHHT. Moreover, to unify chatter indexes for different cutting depths of cut *a_p_*, the ICI are divided by *a_p_*. 

## 4. Nonlinear Time Series Analysis

In the next two subsections, the analytical stability lobe diagram is verified by the use of methods presented in Section 3. For this purpose, the resultant force *F_xy_* is taken into consideration, as a representative of milling dynamics. 

### 4.1. Improved Hilbert–Huang Transform

The DHHT method was used to extract frequency components from the resultant *F_xy_* force signal. Based on a preliminary experiment, the amount of desired components, related to chatter vibrations, were isolated. Next, the instantaneous amplitudes, frequencies and amplitude spectra were calculated. In Figure 6 and Figure 7, the first three components are shown for stable and unstable conditions respectively (according to SLD presented in Figure 3). The first IMF component was connected with the higher frequency vibrations of the milling machine as well as with measurement disturbances. The frequency of this component was spread in a wide range of frequencies from about 800 to 2000 Hz. The second component was connected with the tool-holder system, which has a natural frequency of about 500–700 Hz. The third component was probably connected with lower frequency vibrations of the workpiece. Its frequency was about 300 Hz. As may be seen from amplitude spectra, all the three components were separated quite well, although slight energy leaks between components are still visible. For each IMF component statistical features of instantaneous amplitudes and frequencies, as well as maximum amplitudes of component spectra, were calculated. It was found that qualitative changes in these quantities may be found, especially for the second and third IMF components. From Figure 6 and Figure 7, qualitative changes in amplitude and amplitude variance of the second IMF component may be observed.

In Figure 8, time–frequency spectrograms of resultant force *F_xy_* for a constant depth of cut *a_p_* equal to 0.5 mm and two rotational speeds *n* equal to 6000 and 7000 rpm (corresponding to stable and unstable machining) are shown. As can be seen, the energy of the second and third component is spread in a quite wide range of frequencies, which indicate an irregular and chaotic course of the milling process. Additionally, at lower frequencies, below 200 Hz, the IMF components resulting from the cutting tool rotation during material removal are visible. Moreover, a large increase in the energy contained in the second and third components can be seen over time, which may be due to the discontinuous and non-uniform structure of the CFRP. Therefore, a relatively long time series of force signals should be processed to obtain repeatable analysis results. In this work, 2 s segments in length are taken for analysis. 

In order to show the averaged frequency content of the decomposed force components, the DHHT analysis was performed on relatively short data segments, with a width of 1000 sampling points, which corresponds to 0.25 s of milling. These segments were taken sequentially from a 10 s time series of resultant cutting force. Then, the average instantaneous amplitude and frequency for each IMF component were determined for each segment. The results of this analysis for stable and unstable conditions are presented in Figure 9 and Figure 10. In Figure 9, diagrams for two rotational speeds are shown for stable and unstable machining.

As may be seen, the individual IMF components can be more easily recognized and analyzed. Especially for the second IMF component, a significant increase in amplitude is seen, being a symptom of chatter vibrations. In Figure 10, two spectrograms are shown for constant rotational speed and two depths of cut equal to 0.5 and 1.5 mm. Again, a big increase in the amplitude of the second IMF component was observed. Moreover, a mean frequency of the second IMF component slightly decreases for unstable machining conditions from about 650 Hz to about 550 Hz. This may be due to the increase in damping in the cutting zone caused by the increase of the contact area on the flank surface of the cutter [67].

Summing up, the application of the improved HHT transform applied for the resultant force signals (*F_xy_*) is useful for detecting chatter in the cutting process. The second component of the DHHT seems to be responsible for chatter vibrations because its frequency is between 400 and 700 Hz. The maximum amplitude of the second component spectrum is presented in Figure 11 together with the theoretical SLD (obtained by CutPro9, thin blue line), stable (green) and unstable (red) points, proposed here and defined by means of the second component amplitude. A critical value of *A_DHHT_* is marked as a green broken line. The milling process is stable, without chatter vibrations when the amplitude of the second component is smaller than *A_DHHT_*_cr_. On the base of *A_DHHT_*, a new hypothetical SLD (bold blue line) was proposed. The critical depth of cut (dashed red line) was not horizontal like in a linear approach, but reveals nonlinear properties as shown in [39,68]. This is probably due to the nonlinear damping in the cutting zone, which increases with the decrease of tool rotational speed, as reported in [67].

### 4.2. Recurrence Quantification Analysis and Chatter Indexes

The largest Lyapunov exponent (Figure 12) is not good enough for any components of DHHT, although the second component seems to be the best. Stable and unstable milling points cannot be recognized properly. Therefore, the newly introduced chatter indexes, defined in Equations (14–18), are proposed. They identify chatter much better as shown in Figure 13, Figure 14, Figure 15, Figure 16 and Figure 17. 

In Figure 13 the Lyapunov chatter index (*L_yap_CI*) with proposed critical limit value (*L_yap_CI*_cr_, green dashed line) is presented. The second component of the DHHT identified chatter the best, while the first one was the least accurate. The *L_max_CI*, shown in Figure 14, also indicates instability on the basis of the second component. The critical index *L_max_CI*_cr_ equals about 7.5. Below the green dashed line cutting process is unstable. The *L_ent_CI* (Figure 15), the *LCI* (Figure 16) and the *TTCI* (Figure 17) had significantly smaller values in case of chatter than for stable milling. Thus, the proposed chatter indexes were precise and much better than typical RQA measures. Moreover, they were consistent with the hypothetical SLD presented by the blue line.

In the case of cutting with different depths of cut, the ICIs are related to cutting depth *a_p_* as mentioned in Section 3.2. Then, the ICIs were much better than pure RQA. For instance, the Lyapunov exponent shown in Figure 18a increases with the cutting depth, but in the last case of *a_p_* = 2 mm, it decreases (the first and second component). Only the third component could be taken as a stability index. However, all the ICIs presented in Figure 18b, Figure 19 and Figure 20 diminish with the depth of cut, showing a transition from stable to unstable cutting. The *L_yap_CI* is much higher when the process is stable and less regular than in case of chatter vibrations. The second component is as good as the third one in the analysis of chatter at growing cutting depth. The proposed critical values of ICIs (dashed green lines) were as follows: *L_yap_CI*_cr_ = 1 (Figure 18b), *L_max_CI*_cr_ = 10 (Figure 19a), *L_ent_CI*_cr_ = 1 (Figure 19b), *LCI*_cr_ = 1 (Figure 20a) and *TTCI*_cr_ = 1 (Figure 20b). Thus, the improved chatter indexes work very well and can be used in practice with success.

## 5. Conclusions

The paper is focused on the assessment of the stability of the CFRP milling process by newly proposed, improved chatter indexes that have not been used before in the literature. The improved Hilbert–Huang transform is the first step in the process stability analysis. The resultant cutting force is decomposed into three components, where the second one is responsible for chatter vibrations in the milling process. The amplitude of the second component indicates some symptoms of instability. However, calculation of the improved chatter indexes as the recurrence quantification measures related to the amplitude of chatter component are much better at recognizing CFRP cutting instability. 

The theoretical stability lobe diagram, created with the help of the commercial software CutPro9, is not good enough for chatter detection of milling of carbon fiber reinforced plastics. This is because CFRP is a strongly nonlinear material having different properties from classical constructional materials defined in a model used in CutPro9. The here-proposed hypothetical SLD with non-constant critical depth of cut is proven by experiment and consistent with the improved chatter indexes.

Summing up, the identification of chatter vibrations with the help of joined and improved HHT and RQA is much better than classical linear analysis. Finally, three new chatter indexes are selected, which identify cutting instability as the best. They are *L_yap_CL*, *LCI* and *TTCI*. Each of them is appropriate both for speed and cutting depth-induced instability. 

To create a process supervision (control) system, the machine tool should be equipped with force or alternatively acceleration sensor and data acquisition system to collect the time-series on-line during the process. Next, a computer could do the analysis of the signal according to the procedure presented in Section 3. Finally, software should compare the chosen chatter indexes (CI) to the reference (critical) values and send information to the controller to change rotational speed if the process is unstable. Chatter appears when the critical CI is exceeded. It is important at the beginning to estimate the critical CIs for the specific material, machine and cutting tool. It is important, also, that the control system is universal and can be applied for different materials, not only for composites. Additionally, some feature-fusion techniques, e.g., principal component analysis, could be helpful in creating a more reliable model for chatter prediction.

## Figures and Tables

**Figure 1 materials-13-04105-f001:**
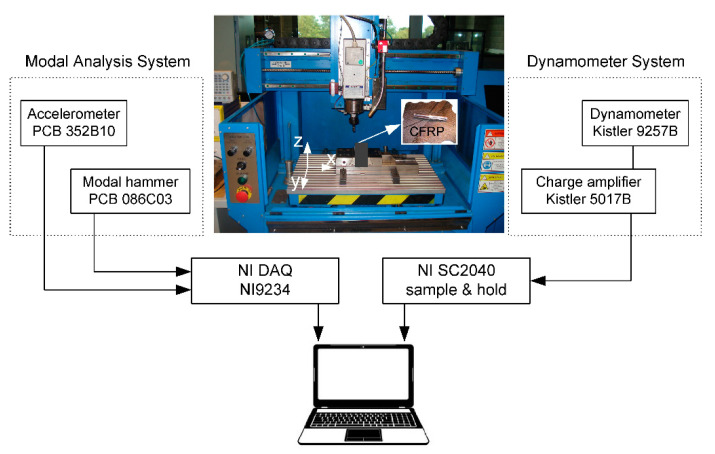
Scheme of experimental setup consisting of milling machine, modal analysis system and dynamometer system.

**Figure 2 materials-13-04105-f002:**
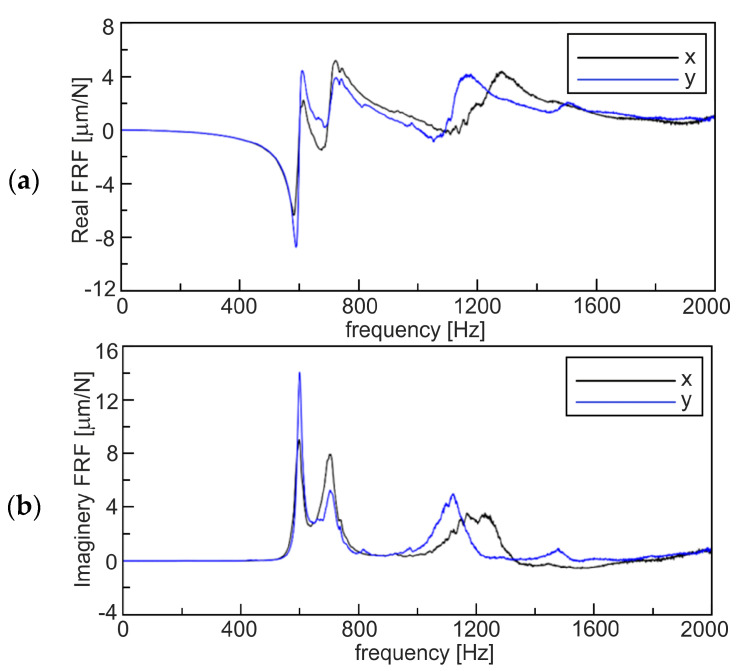
Frequency response function (FRF) of the tool-holder system. Real (**a**) and imaginary part (**b**).

**Figure 3 materials-13-04105-f003:**
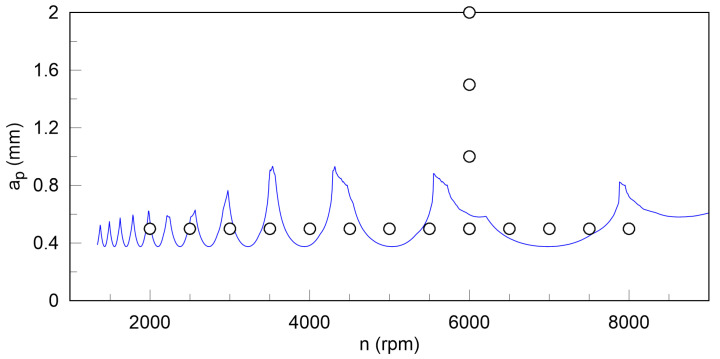
Theoretical stability lobe diagram (SLD, blue line) determined based on CutPro9 software together with proposed points of milling tests.

**Figure 4 materials-13-04105-f004:**
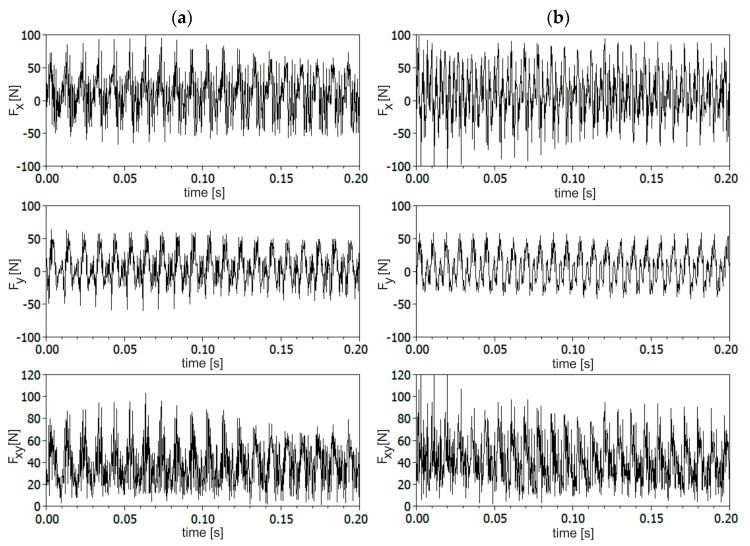
Changes of *F_x_*, *F_y_* components and resultant cutting force *F_xy_* acting on the end mill cutter for: (**a**) stable (*a_p_* = 0.5 mm, *n* = 6000 rpm) and (**b**) unstable cutting (*a_p_* = 0.5 mm, *n* = 7000 rpm).

**Figure 5 materials-13-04105-f005:**
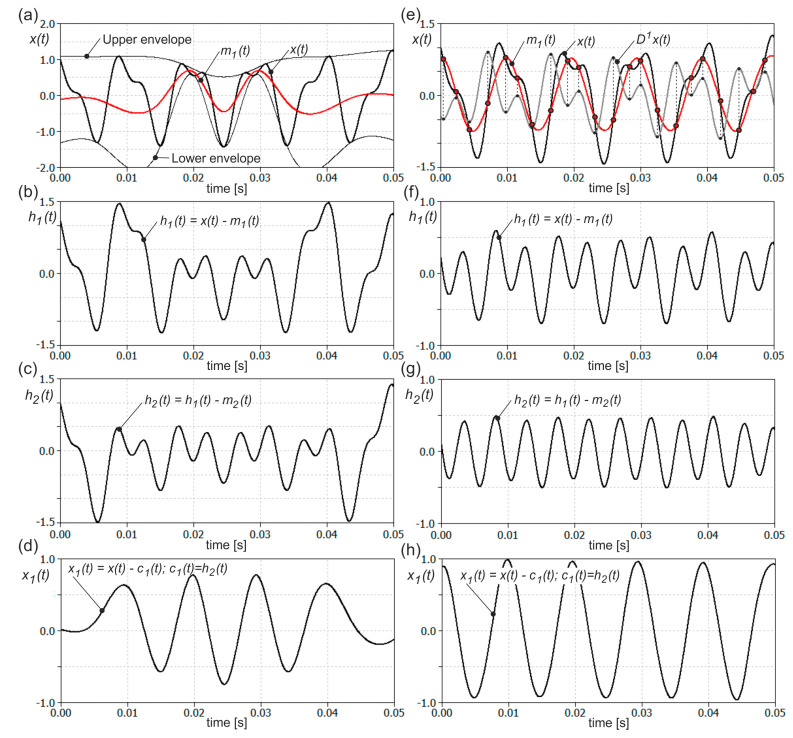
Empirical mode decomposition performed on two-tone signal *x*(*t*) *= cos*(*2π f_1_ t*) − *0.43sin*(*2π f*_2_
*t*) where *f*_1_ = 102 Hz and *f*_2_ = 214 Hz: (**a**–**d**) with the use of standard mean line estimation based on upper and lower envelopes and (**e**–**h**) with the use of mean line estimation based on inflection points derived from the first derivate of the signal.

**Figure 6 materials-13-04105-f006:**
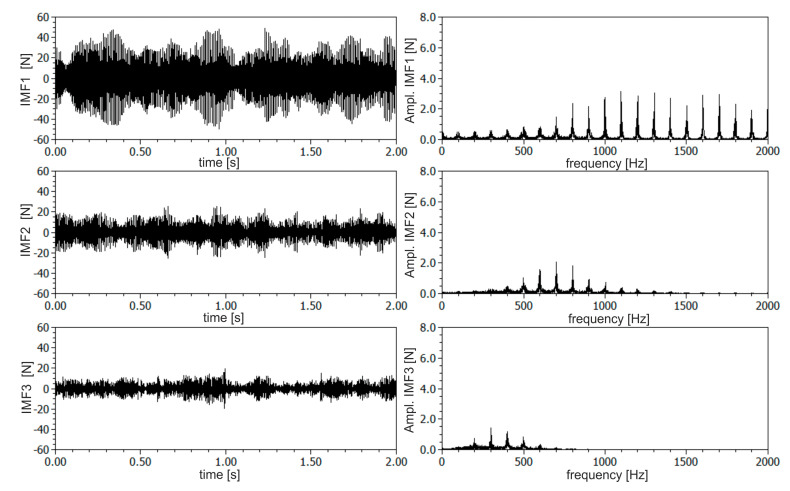
The first three components and their amplitude spectra extracted from the resultant force *F_xy_* for stable machining (based on the theoretical SLD), *a_p_* = 0.5 mm, *n* = 6000 rpm.

**Figure 7 materials-13-04105-f007:**
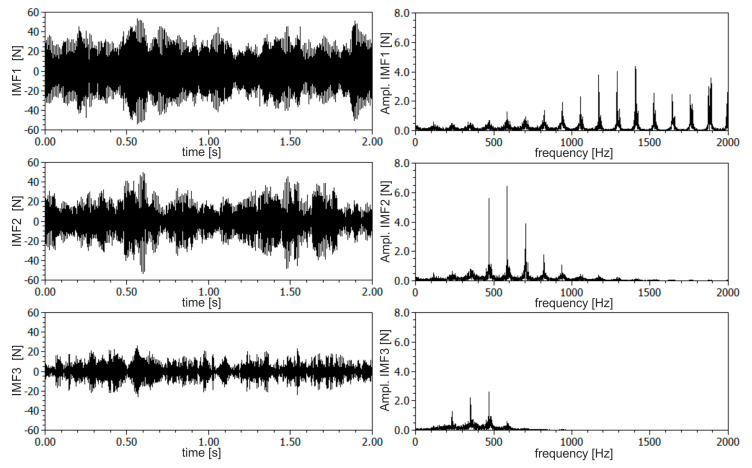
The first three components and their amplitude spectra extracted from the resultant force *F_xy_* for unstable machining (based on the theoretical SLD), *a_p_* = 0.5 mm, *n* = 7000 rpm.

**Figure 8 materials-13-04105-f008:**
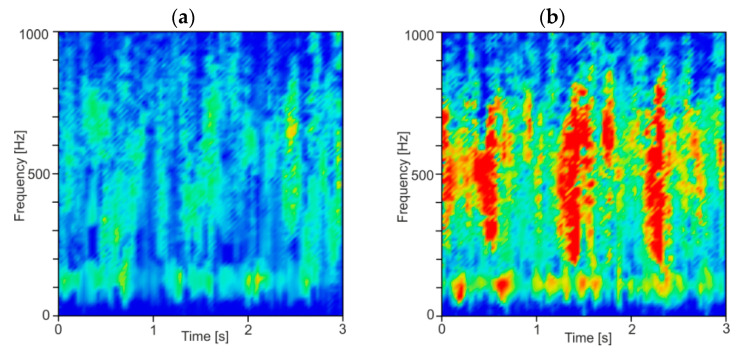
Time–frequency spectrogram for: (**a**) stable (*a_p_* = 0.5 mm, *n* = 6000 rpm) and (**b**) unstable cutting (*a_p_* = 0.5 mm, *n* = 7000 rpm).

**Figure 9 materials-13-04105-f009:**
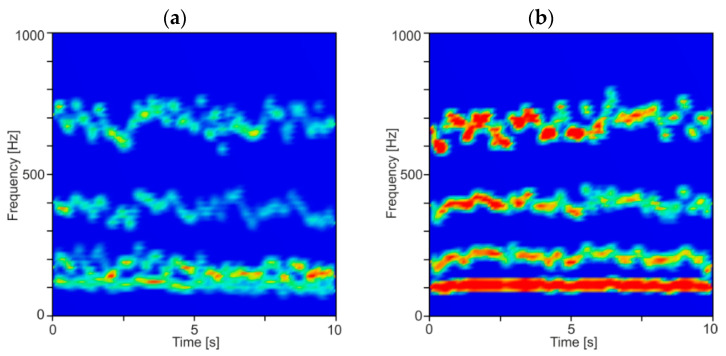
Time–frequency spectrogram for: (**a**) stable (*a_p_* = 0.5 mm, *n* = 8000 rpm) and (**b**) unstable cutting (*a_p_* = 0.5 mm, *n* = 7000 rpm).

**Figure 10 materials-13-04105-f010:**
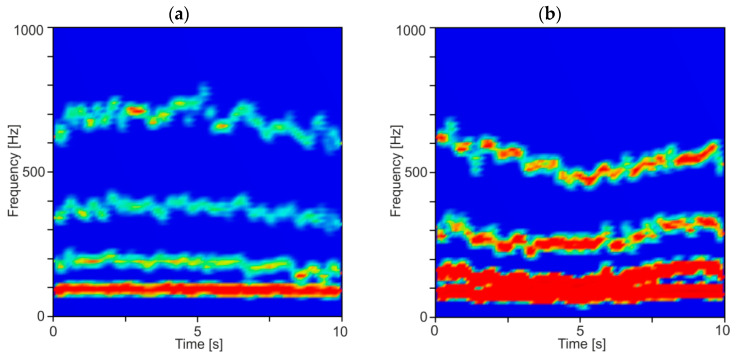
Time–frequency spectrogram for: (**a**) stable (*a_p_* = 0.5 mm, *n* = 6000 rpm) and (**b**) unstable cutting (*a_p_* = 1.5 mm, *n* = 6000 rpm).

**Figure 11 materials-13-04105-f011:**
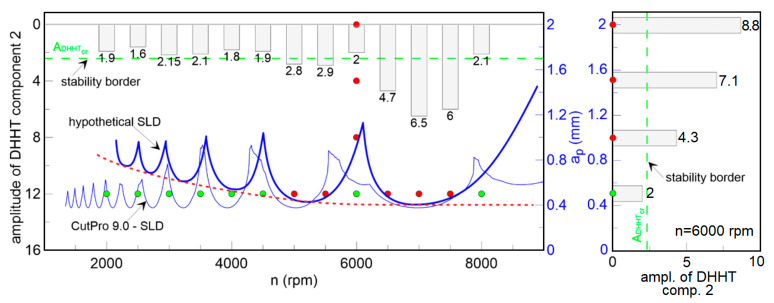
The amplitude of DHHT component 2 (left axis) and the stability lobes diagram (right axis) as a function of rotational speed and depth of cut.

**Figure 12 materials-13-04105-f012:**
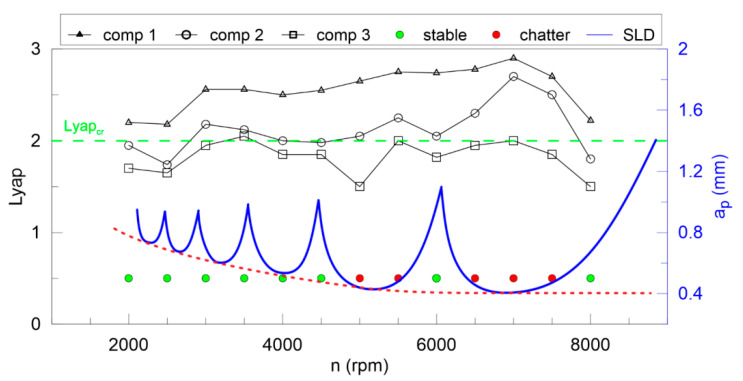
The largest Lyapunov exponent (left axis) and the hypothetical stability lobe diagram (right axis) versus rotational speed.

**Figure 13 materials-13-04105-f013:**
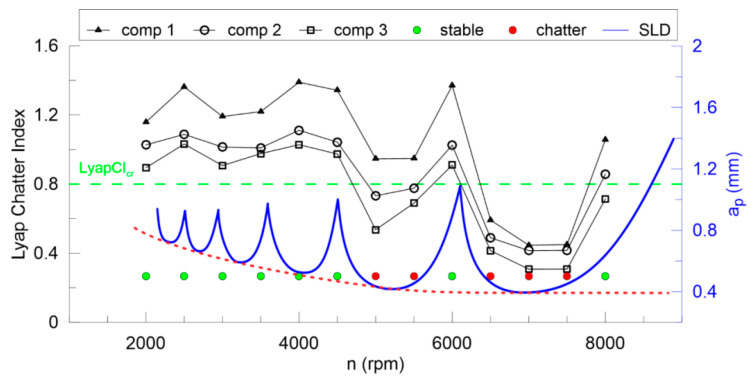
The largest Lyapunov chatter index (left axis) and the hypothetical stability lobe diagram (right axis) versus cutting velocity.

**Figure 14 materials-13-04105-f014:**
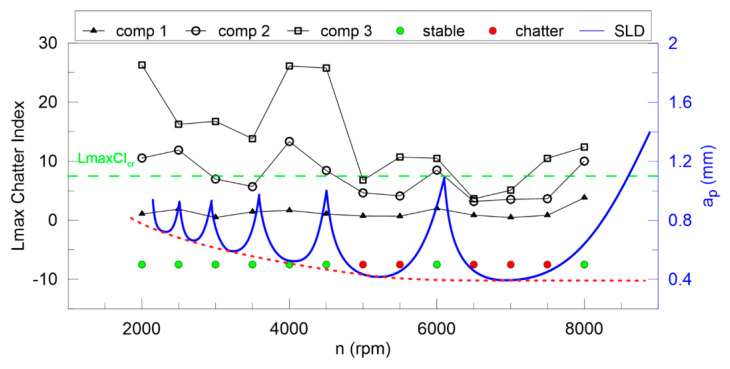
The *L_max_* chatter index (left axis) and the hypothetical stability lobe diagram (right axis) versus cutting velocity.

**Figure 15 materials-13-04105-f015:**
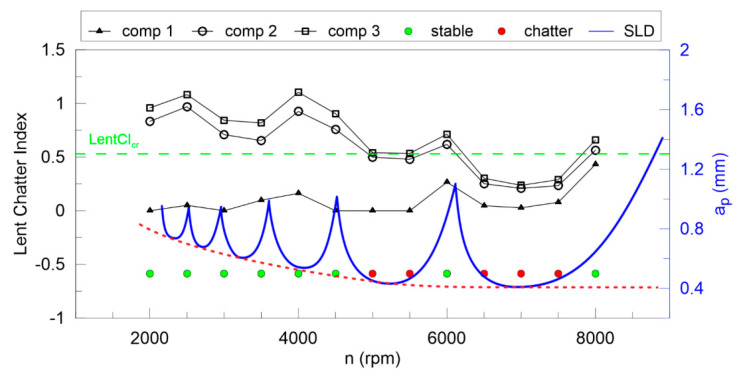
The *L_ent_* chatter index (left axis) and the hypothetical stability lobe diagram (right axis) versus cutting velocity.

**Figure 16 materials-13-04105-f016:**
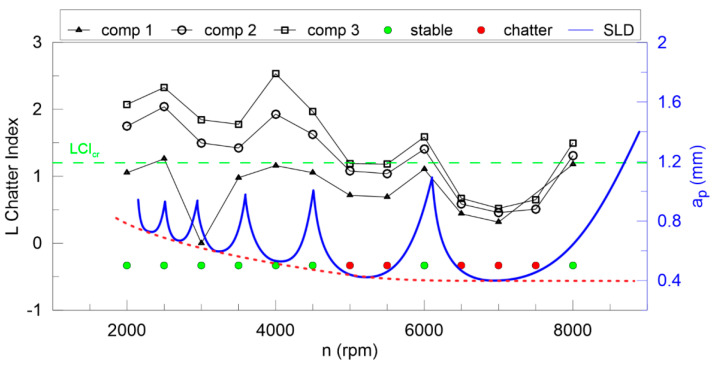
The L chatter index (left axis) and the hypothetical stability lobe diagram (right axis) versus cutting velocity.

**Figure 17 materials-13-04105-f017:**
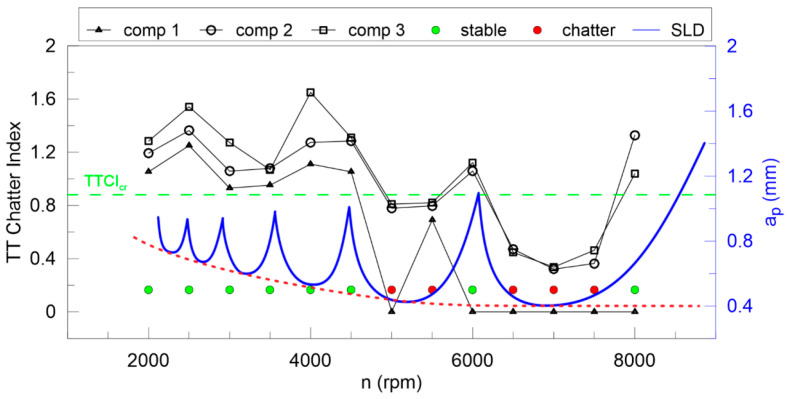
The TT chatter index (left axis) and the hypothetical stability lobe diagram (right axis) versus cutting velocity.

**Figure 18 materials-13-04105-f018:**
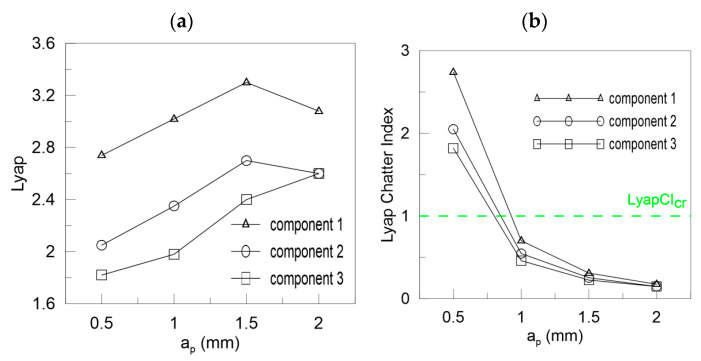
Chatter indexes for *F_xy_* versus depth of cut: (**a**) the largest Lyapunov exponent and (**b**) the Lyapunov chatter index.

**Figure 19 materials-13-04105-f019:**
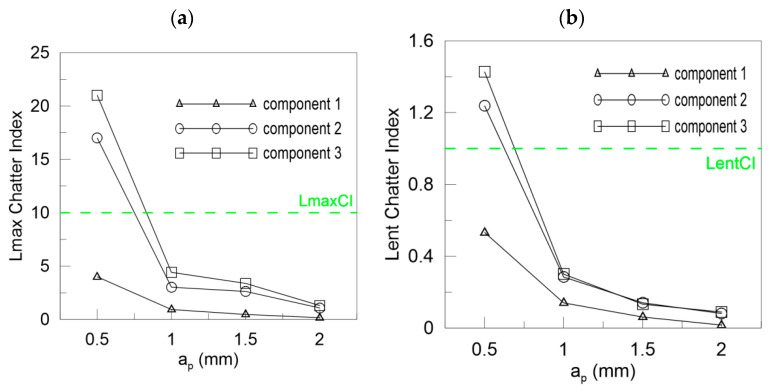
Chatter indexes for *F_xy_* versus depth of cut: (**a**) the *L_max_* and (**b**) *L_ent_* chatter index.

**Figure 20 materials-13-04105-f020:**
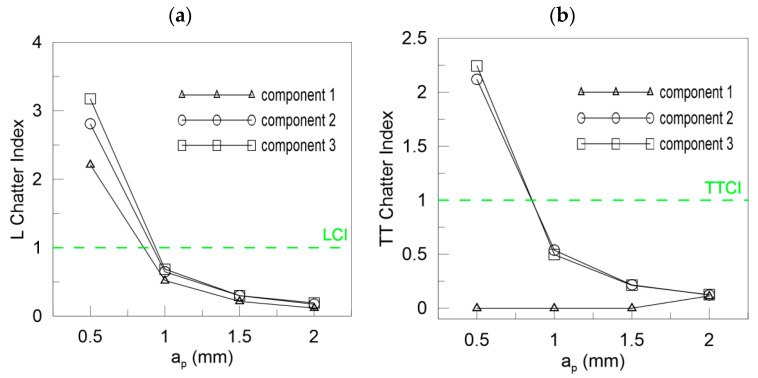
Chatter indexes for *F_xy_* versus depth of cut: (**a**) the L chatter and (**b**) TT chatter index.

**Table 1 materials-13-04105-t001:** Range of parameters used in experimental tests.

Test of rational speed (constant *a_p_*): *a_e_* = 12 mm, *f_z_* = 0.05 mm/flute, *a_p_* = 0.5 mm													
*n*	2000	2500	3000	3500	4000	4500	5000	5500	6000	6500	7000	7500	8000
(rpm)													
Test of cutting depth (constant *n*): *a_e_* = 12 mm, *f_z_* = 0.05 mm/flute, *n* = 6000 rpm													
*a_p_*	0.5	1.0	1.5	2.0									
(mm)													

*a_p_*—cutting depth, a_e_—radial depth of cut, f_z_—feed rate per tooth, *n*—rotational speed.

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
