# Peer review of "Chatter Detection in Milling of Carbon Fiber-Reinforced Composites by Improved Hilbert–Huang Transform and Recurrence Quantification Analysis"

_materials, 2020, doi:10.3390/ma13184105_

Round 1

Reviewer 1 Report

This paper presents a new methodology (using a HH transform and recurrence analysis) to detect chatter in milling of CFRCs. This is a method like many others, for example:

Zhang, Q., Tu, X., Li, F., Hu, Y.  An effective chatter detection method in the milling process using morphological empirical wavelet transform (2020) IEEE Transactions on Instrumentation and Measurement, 69 (8)

Cherukuri, H., Perez-Bernabeu, E., Selles, M., Schmitz, T.  Machining chatter prediction using a data learning model (2019) Journal of Manufacturing and Materials Processing, 3 (2), art. no. 45,

M.-Q., Liu, M.-K., Tran, Q.-V. Milling chatter detection using scalogram and deep convolutional neural network (2020) International Journal of Advanced Manufacturing Technology, 107 (3-4), pp. 1505-1516.

I think the authors should cite these papers and explain why their methodology detects chatter in CFRCs pieces better than the above methods. Their novelty is not clear, since there is no statistical analysis of the accuracy or error of the presented approach. 

In the introduction section, the authors state that  "process supervision systems are crucial, which could diagnose the process state in real-time and accurately modify the machining conditions when some symptoms of process instability appear". How can your methodology be applied in real-time machining? The response should be added to the manuscript. Can be used for self-aware CNC milling machines?

Have you tried with another tool with a different diameter or number of flutes? Does the chatter change if another tool is used? The authors should clarify if their approach is only valid for a specific holder-tool system.

The experimental section shows that n and ap are used as variables, with different levels, but the authors don't show the final DOE design. How many experiments did they realize?

The conclusions section doesn't show which indexes have been chosen at the end (largest Lyapunov exponent, Lyapunov chatter, Lmax, Lent chatter, Lchatter, TT chatter). 

Can your detection methodology be useful for other materials? Please, clarify this in the manuscript.

See the attached PDF document and address my comments on it. 

Author Response

Authors would like to thank the reviewer very much for the important feedback that helped to improve the quality of the paper. All changes, suggested by the reviewer, are marked in red colour in the manuscript.

Responses on the comments are made in Word  file and additionally, answers and our comments are done in PDF as responses for the remarks put in  peer-review-8210668.v2

Reviewer 2 Report

The manuscript provides a very good overall expression. It is not only well written and informative, also the data presented are of high interest and novelty. The chatter vibration occuring during CFRP milling process may reduce the efficiency of the process and therefore require an accurate prediction techniques. The manuscript aims at resolving this issue by providing a reliable model for chatter prediction based on cutting force component measurements.

At first, a literature review on chatter vibration is given that describes the current state in this area. Moreover, the experimental background presented in a manuscript is strong and does not require any improvements. The data for experimentally measured cutting forces went through HHT; Statistical evaluations and RQA were applied to detect chatters. As a result, new chatter indexes obtained have an obvious practical value and can be further used for the method optimisation.

I would recommend the manuscript for publication in its current state.

Author Response

Authors would like to thank the reviewer very much for the important and nice feedback.

Reviewer 3 Report

Dear Authors,

The presented paper describes an important aspect of composite materials milling as one of processing stages. Milling as an energy demanding process requires a special attention in process efficiency control and performance. Thus, the process stability evaluation is critical for continuous applications. In a whole, the paper adds a deeper knowledge to vibration analysis associated with milling processes

Author Response

(The authors gave the same response as above.)

Reviewer 4 Report

In the paper the problem of chatter vibration detection in the milling process of carbon fiber reinforced plastic is investigated. First, a stability lobe diagram is created to establish the range of experimental test in the plane of tool rotational speed and depth of cut. Next, an experiment of composite milling is performed. The measured experimentally time series of cutting forces are decomposed with the use of improved Hilbert-Huang transform. To detect chatter, the indexes, derived directly from the HHT and RQA methods, can be used to build effective chatter prediction system. Besides, the paper is well organized and expounded clearly. It’s suggestted that this manuscript can be accepted directly.

Author Response

Authors would like to thank the reviewer very much for the important and nice feedback. In the reviewed manuscript English is improved by the authors.